# Integrated Approaches to Reveal Genes Crucial for Tannin Degradation in *Aureobasidium melanogenum* T9

**DOI:** 10.3390/biom9090439

**Published:** 2019-09-02

**Authors:** Lin-Lin Zhang, Jie Li, Yi-Lin Wang, Song Liu, Zhi-Peng Wang, Xin-Jun Yu

**Affiliations:** 1College of Chemistry & Environmental Engineering, Shandong University of Science & Technology, Qingdao 266510, China; 2Laboratory for Marine Fisheries and Aquaculture-Qingdao National Laboratory for Marine Science and Technology, Yellow Sea Fisheries Research Institute, Chinese Academy of Fishery Sciences, Qingdao 266071, China; 3College of Science, China University of Petroleum, Qingdao 266580, China; 4Development & Reform Bureau, West Coast New Area, Qingdao 266000, China; 5Key Laboratory of Sustainable Development of Polar Fishery, Ministry of Agriculture and Rural Affairs, Yellow Sea Fisheries Research Institute, Chinese Academy of Fishery Sciences, Qingdao 266071, China; 6Key Laboratory of Bioorganic Synthesis of Zhejiang Province, College of Biotechnology and Bioengineering, Zhejiang University of Technology, Hangzhou 310014, China

**Keywords:** tannins biodegradation, *Aureobasidium melanogenum*, Tannase, gallic acid decarboxylase

## Abstract

Tannins biodegradation by a microorganism is one of the most efficient ways to produce bioproducts of high value. However, the mechanism of tannins biodegradation by yeast has been little explored. In this study, *Aureobasidium melanogenum* T9 isolated from red wine starter showed the ability for tannins degradation and had its highest biomass when the initial tannic acid concentration was 20 g/L. Furthermore, the genes involved in the tannin degradation process were analyzed. Genes *tan A*, *tan B* and *tan C* encoding three different tannases respectively were identified in the *A. melanogenum* T9. Among these genes, *tan A* and *tan B* can be induced by tannin acid simultaneously at both gene transcription and protein expression levels. Our assay result showed that the deletion of *tanA* and *tanB* resulted in tannase activity decline with 51.3 ± 4.1 and 64.1 ± 1.9 U/mL, respectively, which is much lower than that of *A. melanogenum* T9 with 91.3 ± 5.8 U/mL. In addition, another gene coding gallic acid decarboxylase (*gad*) was knocked out to better clarify its function. Mutant *Δgad* completely lost gallic acid decarboxylase activity and no pyrogallic acid was seen during the entire cultivation process, confirming that there was a sole gene encoding decarboxylase in the *A. melanogenum* T9. These results demonstrated that *tanA*, *tanB* and *gad* were crucial for tannin degradation and provided new insights for the mechanism of tannins biodegradation by yeast. This finding showed that *A. melanogenum* has potential in the production of tannase and metabolites, such as gall acid and pyrogallol.

## 1. Introduction

*Aureobasidium* genus is popularly known as black yeast due to its melanin production and is frequently encountered in soil, water, the phylloplane, wood, and many other plant materials [1,2]. Their ubiquitous distribution in diverse environmental conditions makes them an easily accessible source for biotechnological and environmental applications [3]. Thus, *A. pullulans* has been regarded as a safe and important biotechnological yeast for bioproducts’ production, like poly (β-l-malic acid), heavy-oil liamocins, amylase, proteinase, lipase, pullulan, and siderophore [1,4,5,6,7,8].

Tannins are the second most abundant group of phenols in nature, and their molecular weight depends on the bonds possessed with proteins and polysaccharides [9]. Nevertheless, owing to the ability to form strong complexes with different minerals and macromolecules, tannins not only aggregate the precipitates in the beverages but also raise serious environmental pollution problems [9,10,11]. In addition, tannins negatively affect the nutritional quality of the feed and significantly reduce the intake by livestock because of their bitter taste [12,13].

These undesirable effects of tannins can be reduced or eliminated by certain microorganisms or enzymes. The degradation pathway of hydrolyzable tannins, like gallotannins, has been well understood in bacterial and fungal systems. In the fungal strain, the first step is degradation of tannic acid to form glucose and gallic acid which is converted to pyrogallol by gallic acid decarboxylase. Later, pyrogallol is converted to hydroxymuconic acid by pyrogallol dioxygenase, and then is converted to pyruvate. Pyruvate is finally metabolized through tricarboxylic acid cycle [13]. However, compared to bacteria and fungi, only a few genera in yeast, like *Arxula adeninivorans*, *Candida utilis*, *Debaryomyces hansenii*, and *Mycotorula japonica*, were observed to degrade the tannins, and little is known about their tannin-degrading pathways [12,14,15,16]. To the best of our knowledge, *A. melanogenum* T9 was the first strain to be identified as capable of tannin degrading in the *Aureobasidium* genus.

Tannase is one of the most studied enzymes in tannins biodegradation, and it can break the ester bonds present in gallotannins, ellagitannins, complex tannins, and gallic acid esters. Tannase have some important applications in various industries and commercially available tannases have mainly been produced by the fermentation of fungi up to now, especially the *Aspergillus* species [17]. Many strains of *A. pullulans* also harbor a wide range of industrially important enzymes due to their ubiquitous occurrence, biochemical diversity and easily accessible features. Therefore, tannase production by *Aureobasidium* genus might constitute an adequate alternative to fungal enzymes [3]. It has been found that several different tannases coexist in the fungal or yeast strain, like *Colletotrichum graminicola* M1.001, *Aspergillus niger* CBS513.88 and *Aureobasidium melanogenum* CBS 110374 [18]. Up to now, the strongest reviews regarding tannases were focused on industrial applications and patentability; the respective roles of these genes in tannin catabolism has not been analyzed.

Gallic acid decarboxylase is a kind of nonoxidative aromatic acid decarboxylase and induced by tannins or gallic acid. It is responsible for the decarboxylation of gallic acid to pyrogallol, the second step in the degradation of polyphenol tannic acid. This phenol derivate pyrogallol also has several important industrial applications, for example, in staining of leathers, developing agent in photography, coloring of hair, and cosmetic products [19,20]. Recently, the function of gene *AGDC1* coding gallic acid decarboxylase was studied in order to elucidate its role in tannic acid degradation [14]. Interestingly, gene *AGDC1* deletion leads to growth inhibition of *A. adeninivorans* when gallic acid is present in culture medium. However, it is still not known if this phenomenon also occurs in other yeast strains. Therefore, the function of gene-coding gallic acid decarboxylase in the other strain need to be investigated.

In the present study, a yeast strain of *A. melanogenum* T9 isolated from red wine starter showed an ability for tannins degradation and its tannin tolerance was studied. Three different coding-tannase genes *tan A*, *tan B* and *tan C* and *gad* gene-encoded gallic acid decarboxylase, respectively, were determined and analyzed by bioinformatics method. Our result showed that *tan A*, *tan B* and *gad* gene expression level were significantly induced by tannin acid, thus the function of these genes was analyzed by gene knockout method in order to elucidate their role in tannic acid catabolism.

## 2. Materials and Methods

### 2.1. Regents and Instruments

Tanic acid, gallic acid, pyrogallic acid, yeast extract, tryptone, glucose, NH_4_NO_3_, NaCl, bromophenol blue, agar, and other reagents used in this study were purchased from Sangon company (Shanghai, China). Hygromycin and driselase were purchased from Solarbio company (Beijing, China). Primers were synthesized by Synbio Technologies corporation (Suzhou, China). PCR master mix and SYBR real-time PCR master mix were purchased from Thermo Scientific (Sunnyvale, CA, USA).

Microorganism cultivation was performed in the incubator and shaker of Zhichu company (Shanghai, China). qRT-PCR assay was carried out by ABI 7500 (Applied Biosystems, Arlington, VA, USA).

### 2.2. Strains, Plasmids, and Media

The strain T9 originally from red wine starter was used throughout this study and routinely grown in YPD medium containing 1.0% yeast extract, 2.0% tryptone and 2.0% glucose. *Aureobasidium melanogenum* CBS 110374 was purchased from CBS-KNAW Collections. Tannins degradation test medium with 2.0% tannin, 1.0% NH_4_NO_3_, 0.2% bromophenol blue, and 3.0% agar was employed to determine the tannins degradation ability of the obtained yeast strains. The YPT broth medium used for analyzing the tannin degradation of strain T9 consisted of 1.0% yeast extract, 2.0% tryptone and 2.0% tannins. *Escherichia coli* strains DH5α were used as host strains for bacterial transformation and plasmid isolation. The strains were cultured in Luria Bertani medium supplemented with 100 mg/L ampicillin or 50 mg/L kanamycin if necessary.

### 2.3. Isolation, Phylogenetic Analyses of Tannin-Degrading Yeast Strain

Samples were collected from red wine starter and used as sources of yeast strains. The yeast isolation and screening were performed as previously reported [21]. As for the screening of tannin-utilizing yeast strains, the visual reading method was executed according to Kumar, with some modifications [22]. A single yeast colony was picked and inoculated the plate of the test medium at 28 °C for 3 days, then green to brown coloration of the medium was judged as a positive result for tannin degradation. The strain T9 showing a clear zone around the colony was used for the subsequent investigation, and its phylogenetic relationships with the typical strains reported were analyzed by amplification and sequencing of the internal transcribed spacer (ITS). A phylogenetic tree was constructed by the method of maximum likelihood to determine the phylogenetic position of the strain T9 with MEGA version 6 [23].

### 2.4. Tannic Acid Tolerance Analyses by A. melanogenum T9

The tannin degradation process of *A. melanogenum* T9 strain was analyzed by the estimation of cells biomass, tannase activity, the content of tannic acid, pyrogallic acid, and gallic acid. The *A. melanogenum* T9 yeast cells were inoculated into the YPD medium for the seed culture at 28 °C, and then the seed cultures were carried out in 250 mL flasks containing 50 mL of YPT medium. The solutions of tannins of various concentrations were filter sterilized using a cellulose nitrate membrane of 25 mm diameter and 0.45 mm pore size (Whatman Ltd., Maidstone, UK). These inoculated flasks were incubated at 28 °C with shaking and monitored at the incubation period. Biomass formed in each of the flasks was filtered through and dried at 60 °C overnight. Samples were collected every 12 h during the cultivation. The biomass content was gravimetrically calculated. Tannic acid was determined by the phenol-sulfuric acid method following the modifications reported by Ref. [24]. A reliable quantification of tannase activity was measured by estimating the gallic acid formed due to enzyme action [25]. One tannase unit was defined as the enzyme amount needed to release 1 µmol of gallic acid per min. Gallic acid decarboxylase was detected as Meier described [14]. Three independent assays were carried out and the average values were calculated.

### 2.5. Gene Expression Level Analyses with qRT-PCR Assay

The seed culture of *A. melanogenum* T9 was prepared and a total of 1.0 mL of the seed cultures was inoculated into the YPT medium or YPD medium for 36 h at 28 °C. Afterwards, the yeast cells from the two samples were harvested (10,000× *g* at 4 °C) and resuspended in 10 mM phosphate buffer (PBS, pH 7.2–7.4). Total RNA was isolated using TRIzol reagents described by the manufacturer. RNA concentration was analyzed with NanoDrop2000c spectrophotometer (Thermo Fisher, Bremen, Germany) and reversed transcribed into cDNA. Real-time PCR was performed in triplicate using a SYBR green assay kit (Toyobo, Japan), with the SYBR real-time PCR master mix (Applied Biosystems). The primers for transcription-quantitative PCR (qRT-PCR) were designed according to the sequences of the genes. The primers were listed in Table 1. The 18S rDNA gene was employed as an internal reference [26]. A complete genome sequencing of *A. melanogenum* CBS 110374 has been accomplished and was then selected as reference data for analyses. Three putative genes *tanA*, *tanB* and *tanC* correspondingly coding tannase of TanAp (NCBI access number MN295986), TanBp (NCBI access number MN295987) and TanCp (NCBI access number MN295988) respectively, and gene *gad* coding gallic acid decarboxylase (GAD, NCBI access number MN295989), were found and annotated in *Aureobasidium melanogenum* T9. Relative gene expression level changes were analyzed by using the comparative CT method with a 10-μL reaction system. Samples from YPD medium were used as control. All of the amplifications were performed in triplicate from biological triplicates.

### 2.6. Proteins Expression Level Analyses with Label-Free Technology Mass Spectrometry-Based Label-Free Quantitative Proteomics

Proteins expression-level quantitative analyses of TanAp, TanBp, TanCp and GAD were measured by mass spectrometry-based label-free quantitative proteomic technology. For this purpose, 2 mL cultures from the YPT medium or YPD medium were centrifuged for 10 min at 4 °C with 10,000× *g*. Five-hundred microliters of supernatant was adjusted to pH 2 with HCl, and compounds were extracted with 1 mL MTBS (Sigma-Aldrich, USA). The obtained total proteins of yeast cells were then digested and identified by liquid chromatography (LC)-electron spray ionization (ESI)-tandem mass spectrometry (MS/MS) analyses. For proteins quantitation, these proteins were weighted and normalized relative to the median ratio in NovoGene Corporation (Beijing, China).

### 2.7. Genes Function Analyses by Construction of Mutant Strains

Gene disruptions were performed as described by Chi [27]. According to the sequence of the disruption vector, a fragment containing Poly(A)-hygromycin B phosphotransferase (HPT) gene–TEF promoter was synthesized (Synbio Technologies, China). Primers A5F, A5R, A3F, and A3R were designed according to the coding sequence (CDS) of *tanA*. Primers A5F and A5R were used to amplify the 5′-arm; Primers A3F and A3R were used to amplify the 3′-arm. Primers HPT5 and HPT3 were used to amplify the fragment Poly(A)-hygromycin B phosphotransferase (HPT) gene–TEF promoter. Then, the 5′-arm, 3′-arm, and DNA fragment containing Poly(A)-hygromycin B phosphotransferase (HPT) gene-TEF promoter were connected by overlapping PCR, generating the *tanA*-knocking fragment. This obtained fragment was transformed into the competent cells. The disruptants were grown on the YPD plate containing 50.0 μg/mL hygromycin for 2 days. One colony was picked as strain *ΔtanA*. Primers B5F and B5R were used to amplify the 5′-arm of the *tanB*-knocking fragment; primers B3F and B3R were used to amplify the 3′-arm of the *tanB*-knocking fragment. Primers G5F and G5R were used to amplify the 5′-arm of the *gad*-knocking fragment; primers G3F and G3R were used to amplify the 3′-arm of the *gad*-knocking fragment. *tanB*-knocking fragment and *gad*-knocking fragment were constructed as above; the genes of *tanB* and *gad* were disrupted using the same method. To check the synergistic effect of *tanA* and *tanB*, strain *ΔtanA*, strain *ΔtanB*, strain *Δgad*, and wild strain T9 were prepared, and a total of 1.0 mL of the seed cultures was inoculated into the YPT medium for 36 h at 28 °C. Gene expression level analyses of *tanA* and *tanB* were carried out with qRT-PCR assay as described earlier in the paper. The primers used for the construction of mutant strains are displayed in Table 2.

### 2.8. GAD and Different Tananses Proteins Analyses with Bioinformatics Method

The bioinformatics analyses of TanAp, TanBp, TanCp, and GAD were conducted. Database searches were obtained from NCBI website (https://www.ncbi.nlm.nih.gov/) and performed using BlastX (http://www.ncbi.nlm.nih.gov/BLAST), and multiple sequence alignments were analyzed with DNAman software package (Version 5.2.2, Lynnon Biosoft, Quebec, Canada). Protein analyses with secretion signal peptide prediction was conducted according to the SignalP (version 4.1) program (http://www.cbs.dtu.dk/ services/SignalP/). Physiochemical data were generated from the ProtParam software using ExPASy server (the proteomic server of Swiss Institute of Bioinformatics). A phylogenetic tree of TanAp, TanBp and TanCp as well as other putative tannase protein sequences from fungi and yeast were developed using the Neighbor-Joining method.

### 2.9. Statistical Analyses

The data were the average of three independent experiments, and the error bars indicated the standard deviations (SDs) from the mean of triplicates. The significant differences among groups were calculated with analyses of variance (ANOVAs) followed by Dunnett’s tests by using GraphPad Prism 5 software (San Diego, CA, USA). *p* values of 0.05 were defined as statistically significant in our study.

## 3. Result

### 3.1. A. melanogenum T9 Having the Ability of Tannic Acid Degradation

After isolation and purification of the yeast strains from the red wine starter samples, it was found that the morphologies of 20 obtained yeast strains were similar to those of *Aureobasidium* spp. The 20 strains were cultivated in tannins degradation test medium and evaluated by their abilities to degrade tannins. Nine strains grew in the medium, indicating that these strains have the ability of tannic acid degradation. Compared with other strains, T9 has great potential for tannins degradation (Appendix A). Also, there was an obvious change from blue to yellow in the test medium (data not shown), which indicated that tannic acid was transformed into a more acidic substance. The colonies of the strain T9 grown on the potato dextrose solid medium had yeast-like cells and the colonies are observed as cream, exhibiting characteristics of *Aureobasidium* spp. Furthermore, ITS sequencing and phylogenetic tree construction of the strain T9 were performed as described in “Materials and methods”. According to the high homology (99% identity) with *A. pullulans var. melanogenum* CBS109800, confirmed by topology of the phylograms, the strain T9 was assigned to *A. melanogenum* T9 (data not shown). The genome of the type of strain of this species, *A. melanogenum* CBS 110374, has been sequenced, providing convenience for our study. *A. melanogenum* CBS 110374 also has the ability for tannins degradation, but obviously weaker than strain T9 (Appendix A).

### 3.2. Analyses of Tannin Tolerance by A. melanogenum T9

For the tannin tolerance assay, the *A. melanogenum* T9 strains were cultivated in the YPT medium with different concentrations of tannin (10, 15, 20, 25, and 30 g/L). Figure 1a showed that the biomass of *A. melanogenum* T9 increased with the tannin’s concentration rising and peaked off early with 6.7 ± 0.3 g/L under the condition of 20 g/L. Nevertheless, the growth inhibition was observed when the tannin’s content was beyond 20 g/L, implying that the excess tannins had adverse effects for *A. melanogenum* T9 growth. Furthermore, with the increase of tannin concentration in the medium, the gallic acid content had a clear rising trend, showing that tannic acid was degraded into gallic acid by *A. melanogenum* T9.

### 3.3. A. melanogenum T9 Growth Process Analyses with Tannic Acid as the Sole Carbon Course

The results in Figure 1b showed that the biomass and tannase activity of *A. melanogenum* T9 reached a relative stable level at 48 h. Tannins had an obvious content decline and were totally decomposed after 40-h culture. Meanwhile, with the decrease of tannic acid content, the yield of gallic and pyrogallic acid increased obviously and reached the highest values at 36 h, with 3.1 ± 0.1 g/L and 2.8 ± 0.1 g/L, respectively.

### 3.4. Bioinformatics Analyses of Tannases and GAD

In the *A. melanogenum* T9, tannase of TanAp, TanBp and TanCp contained 528, 587 and 508 amino acids respectively, with predicted protein molecular masses of 57.2, 63.4 and 54.7 kDa accordingly. Proteins belonging to three different tannases shared 18.14% amino acids identity. The multiple sequence alignment of these three tannase protein sequences showed several putative conserved domains, pentapeptide motif Gly-X-Ser-X-Gly typical of serine hydrolases and (acetyl esterase/lipase) and Abhydrolase_3 (alpha/beta hydro-lase fold). The phylogenic tree showed that although the three different tannases had common roots, they diverged in three branches within which they evolved along discrete lines to form subclasses of specialized enzymes (Figure 2a). Therefore, it seemed that TanAp, TanBp and TanCp could be classified into three different tannase enzymes and these proteins’ biochemical characteristics are predicted in Table 3. The data showed that the signal peptide appeared in three different tannases and their evolutionary relationship closed proteins. Though the molecule weight can be used to distinguish TanAp-like tannases (molecule weight from 55.15 to 57.23 kDa) and TanBp-like tannases (molecule weight from 62.57 to 65.65 kDa), it was not suitable to differentiate TanCp-like (molecule weight from 43.54 to 81.47 kDa) from other types of tannases.

The putative GAD protein had an open reading frame of 696 bp, encoding a protein with 232 amino acids. The predicted molecular mass of GAD is 27.3 kDa, and protein analyses revealed the absence of a secretion signal sequence. The phylogenetic tree (Figure 2b) inferred from 1000 bootstrap replicates is taken to represent the evolutionary relationship between GAD and other putative gallic acid decarboxylase protein sequences from yeast and fungi. These proteins, including hypothetical protein (THV69751.1), hypothetical protein (KEQ79373.1), hypothetical protein (THV90437.1), and gallate decarboxylase (SJN60119.1), had a 71.56% similarity of protein sequences. The proposed putative enzymes of *A. pullulans* originated from the same ancestral node, which shares a common ancestor with GAD. The conclusion is that GAD has a closely-related evolutionary relationship with gallic acid decarboxylases from *A. pullulans* and *Blastobotrys adeninivorans*.

### 3.5. Tannic Acid Induced Related Genes Expression Up-Regulation

In Table 4, qRT-PCR data showed that the gene relative expressions of *tanA* and *tanB* cultured in YPT medium were 32-fold and 64.70-fold increased respectively compared to those cultured in YPD medium. Furthermore, the data of Mass Spectrometry-Based Label-Free Quantitative Proteomics demonstrated that these proteins’ expression levels of TanAp and TanBp were 8.22-fold and 332.00-fold higher than those of the control. However, no obvious transcriptional level difference of the *tanC* was observed when the *A. melanogenum* T9 was exposed to tannins. Besides, *gad* also showed a 3.21-fold up-regulation change as compared to that of the control.

### 3.6. tanA and tanB Having Sililar Function on the Tannic Acid Metabolizing

The above-mentioned assay results showed that genes *tanA*, *tanB* and *gad* could be induced by tannic acid. Thus, their role in tannin acid degradation were further analyzed. As shown in Figure 3e, the tannase activity of *A. melanogenum* T9 achieved stability at 48 h with 91.3 ± 5.8 U/mL, which was much higher than that of *ΔtanA* and *ΔtanB* with 51.3 ± 4.1 and 64.1 ± 1.9 U/mL, respectively. On account of tannic acid being used as the sole carbon course in the medium, the tannic acid utilization ratio would have a direct impact on the growth and reproduction of yeast cells. Thus, the tannin degradation and biomass accumulation rate declined, especially within the first 30 h of culture (Figure 3a,b), which was exclusively due to tannase activity reduction in the mutants *ΔtanA* and *ΔtanB*. Even so, mutants *ΔtanA, ΔtanB* and wild-type degraded all of tannic acid at 72 h, and their biomass finally reached the same level at 60 h.

Furthermore, the concentration analyses of gallic acid and pyrogallic acid were performed on the yeast cells of the *ΔtanA, ΔtanB* and wild-type grown for 84 h (Figure 3c,d). It could be seen that the appearance time of maximum concentration of gallic acid in both strains *ΔtanA* and *ΔtanB* was 48 h, which was significantly delayed when compared to wild-type with 36 h. Meanwhile, the gallic acid maximum concentration of *ΔtanA* and *ΔtanB* was also lower than the wild-type strain. The appearance time delay and decline of maximum gallic acid concentration in *ΔtanA* and *ΔtanB* may partly be attributed to the reduction of tannase activity and tannic acid degradation rate. Moreover, the relative transcription level of gene *tanB* in *ΔtanA* increased to 183.2%, compared with that in the wild strain; the relative transcription level of gene *tanA* in *ΔtanB* increased to 121.2%, compared with that in the wild strain (Appendix A). In the *Δgad* strain, the relative transcription level of both genes of *tanA* and *tanB* decreased significantly (Appendix A).

### 3.7. Gad Was Crucial for on the Tannic Acid Metabolizing and Growth

In Figure 3d,f, our experiment result showed that no release of pyrogallic acid and gallic acid decarboxylase activity was seen during the entire cultivation process in the mutant *Δgad* compared to the parental strain. As gallic acid decarboxylase could catalyze gallic acid into pyrogallic acid, it can be easily concluded that there was only one gene encoding decarboxylase in the *A. melanogenum* T9. The decarboxylase activity disappearing further resulted in the excessive accumulation of gallic acid within the first 60 h of incubation. However, the accumulative gallic acid was finally almost completely degraded by *Δgad*, suggesting that gallic acid was utilized by *A. melanogenum* T9 by other unknown metabolic pathways (Figure 3c). The quantitative analyses data showed that an overall 39.7% reduction in the biomass was found for the mutant *Δgad* compared with *A. melanogenum* T9 after a 72-h culture was conducted (Figure 3b). Additionally, it was worth noting that the gene *AGDC1* deletion also led to the reduction of tannase activity and tannic acid degradation rate (Figure 3a,e).

## 4. Discussion

Tannins are generally considered recalcitrant to biodegradation and have toxic effects on various organisms. However, some microorganisms have evolved to use tannins as carbon and energy sources. Up to now, most of the microorganism species used for tannin biodegradation were determined as fungi or bacteria, few researches were related with yeast. In this study, a tannin-degrading yeast strain of *Aureobasidium melanogenum* T9 was isolated and identified from the red wine starter. *A. melanogenum* T9 was able to grow well under conditions of different contents of tannin and it had the highest biomass when an initial tannic acid concentration of 20 g/L was used (Figure 1b). Bi Shi reported that tannin tolerances of *C. utilis* was adapted to 25 g dm^−3^ by gradually increasing the concentrations of valonea tannins in the culture media, and the most suitable concentration of valonea tannins in liquid fermentation medium for ellagic acid production by *C. utilis* was 9.0 g dm^−3^ [15].

Multiple sequence alignment of TanAp, TanBp and TanCp protein sequences displayed the putative conserved domains of Gly-X-Ser-X-Gly and Abhydrolase_3. The conserved signature sequence “Gly-X-Ser-X-Gly” of motif was unprecedented in serine hydrolases and showed the highest conservancy among all of kinds of bacteria and fungi tannase sequences [28]. The mutational analyses indicated that the disulfide bond between Cys202 and Cys458 is crucial for the activity of AoFaeB [29]. The different types of tannase proteins found in the same yeast or fungal species may result from the cumulative effect of species specificity and varying interaction with external environments. In bacteria, the absence signal peptide as well as molecular size were used to help to identify different subtype tannases; nevertheless, these characteristics were improved for tannases of yeast and fungi. Additionally, the phylogenetic relation between GAD and other similar proteins were identified, and these proteins had a high similarity. In the bacteria, like *Lactobacillus plantarum*, several conserved domains, such as subunit B, were fundamental for enzyme catalytic activity and structure in the Agdc1p of *Lactobacillus plantarum* WCFS1 [30].

Tannase can be induced by phenolic compounds such as gallic acid, pyrogallol, methyl gallate, and tannic acid. However, the induction mechanism has not been demonstrated on the gene expression level [31]. Similar to those of previous researches, our assay result showed that the tannase activity had remarkable rise when tannin was present in the medium. The qRT-PCR assays exhibited the inducibility of gene *tanA* and *tanB*, nevertheless, no significant difference of *tanC* was noticed at the level of gene transcription level in the presence of tannins. It is interesting that the *tanC* gene was not sensitive to the tannins, and whether *tanC* was involved in tannins degradation needs to be explored. These results demonstrated that although the tannase synthesis was induced at the protein expression level by tannin, not every *tan* gene coding tannase was induced by tannin at the gene transcription level. In addition, our assay result showed that tannin also caused the *gad* gene expression level to up-regulate in *A. melanogenum*. Thus, the *tanA*, *tanB* and *gad* gene functions in the tannin degradation process were explored in the subsequent assays.

As expected, the tannase activity had a remarkable decline after gene *tanA* or *tanB* was knocked-out (Figure 3e). The reduction of tannase activity further caused tannin degradation, biomass accumulation rate decreased as well as the appearance time delay, and gallic acid maximum concentration declined (Figure 3a–c). However, mutants *ΔtanA*, *ΔtanB*, and the origin strain did not exhibit biomass differences, and tannins were nearly completely utilized by these strains after an 84-h culture. Of note, *ΔtanA* and *ΔtanB* had a rather high similarity in the biomass accumulative and enzyme activity change trend. It could be concluded that the gene *tanA* performed a similar function in tannin degradation with *tanB*. However, the existence of single *tanA* or *tanB* was enough for the strain to utilize tannins, although causing a growth delay to some extent. No obvious synergistic effect of the two enzymes was observed. These two enzymes may be responsible for catalyzing the same reactions. To further study this, the two enzymes should be expressed and characterized.

The mutation of any of those two genes induces a higher transcription of the other (Appendix A). Double-gene disruption of *tanA* and *tanB* will be carried out in future work, which may cause more significant defects in tannin degradation. The three tannase genes in this study were found based on the sequence similarity and were confirmed by the RNA-seq data and label-free quantitative proteomics. However, in the RNA-seq data, some genes annotated with unknown functions have been found with a certain degree of structural similarity with the identified tannases. In future work, the functions of these genes should be identified.

Additionally, the role of gene *gad* on the tannin degradation was also explored. The quantitative determination of pyrogallic acid content and gallic acid decarboxylase activity from *Δgad* and wild-type (Figure 3d,f) demonstrated that *gad* was the only gene responsible for the gallic acid decarboxylase synthesis in the *A. melanogenum* T9 under this culture condition. Although the *gad* disruption prevented gallic acid effective degradation in some way, the excessive gallic acid was still entirely decomposed by the *A. melanogenum* T9 with other unknown pathways. Usually, gallic acid decarboxylase plays a very important role in the metabolism of tannins and phenol derivatives. However, a recent research exhibited that the deletion of gene *AGDC1* from *A. adeninivorans* also contributed to cells death or growth inhibition when gallic acid was present in culture medium under this culture condition [14]. Similar to a previous report, a growth inhibition phenomenon was noticed in the mutant *Δgad* utilizing the tannic acid as the sole carbon source; how the gene *gad* imposed an adverse impact on the yeast growth remains unclear. It is possible that the excessive gallic acid caused by a lack of gallic acid decarboxylase had a toxic effect on *A. melanogenum* T9, or the gene *gad* played a key role in the yeast cells growth. Thus, the retailed mechanism needs to be explored in the future. In *A. melanogenum* T9, the gene *gad* deletion has been associated with reduction of the tannase activity and tannic acid degradation rate during the tannic acid degradation process. It is possible that the tannase activity decline in the mutant *Δgad* may have resulted from the excessive accumulation of gallic acid. A high concentration of gallic acid may have negative feedback regulation on tannase activity, and tannase activity decline was associated with a concomitant decrease in the tannic acid degradation rate.

Biodegradation by certain microorganisms and enzymes is one of the most efficient ways to degrade large tannin molecules into small molecules with bio-activities of high value, such as gallic acid or pyrogallol [32]. Nowadays, tannase and other metabolites, like gall acid and pyrogallol, secreted by tannin-degrading microorganisms, are gaining more attention because of their industrial importance. Tannase has a long study history over more than a century and an extensive biotechnological application in the field of food stuff detannification and chemical and healthcare product manufacturing [33]. Gallic acid has been used in food, cosmetics, adhesives, manufacture of writing inks and dyes, and the synthesis of trimethoprim in the pharmaceutical industry [34,35]. As the major hydrolytic product of gallic acid, pyrogallic acid was used for leathers staining, hair coloring, photographic plate-developing agent, and anti-tumor drug [19]. On the other hand, the yeast *A. melanogenum* is often used as a host for producing high levels of extracellular proteins and products, since it can grow in simple defined media and reaches very high cell densities. In addition, several studies have shown that the direct use of these strains and their metabolites are safe in the human-related sectors, such as in the medical and food industries [3]. Thus, *A. melanogenum* could be an important candidate for future use in tannin-related industries.

## 5. Conclusions

In this study, several genes crucial for tannin degradation in *Aureobasidium melanogenum* were revealed for the first time by integrated approaches. The optimal tannic acid concentration for *A. melanogenum* T9 growth was supposed to be 20 g/L. The genes encoding tannases and gallic acid decarboxylase were analyzed respectively by gene knock-out method in order to better understand their role during the tannin degradation process. The result revealed that mutants *ΔtanA* and *ΔtanB* had a much lower tannase activity and tannin acid degradation rate than that of the parental strain. Furthermore, no gallic acid decarboxylase activity and pyrogallic acid were detected after gene *gad* was deleted. The gene *gad* knock-out not only led to the drop of tannase activity and the tannic acid degradation rate, but also exerted a negative impact on the growth of *A. melanogenum* T9. The above experimental results provided new insights for the mechanism of tannins biodegradation by yeast and also showed that *A. melanogenum* has the potential for the production of gallic acid and pyrogallol.

## Figures and Tables

**Figure 1 biomolecules-09-00439-f001:**
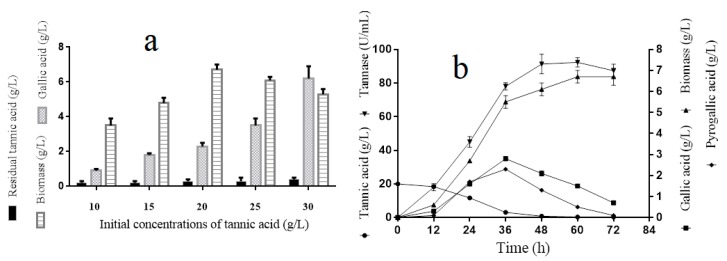
Tannic acid tolerance analyses of *A. melanogenum* T9 under the conditions of different tannin acid concentrations (**a**). Tannic acid degradation process analyses for *A. melanogenum* T9 (**b**). The mean expression values ± SDs are reported relative to the control.

**Figure 2 biomolecules-09-00439-f002:**
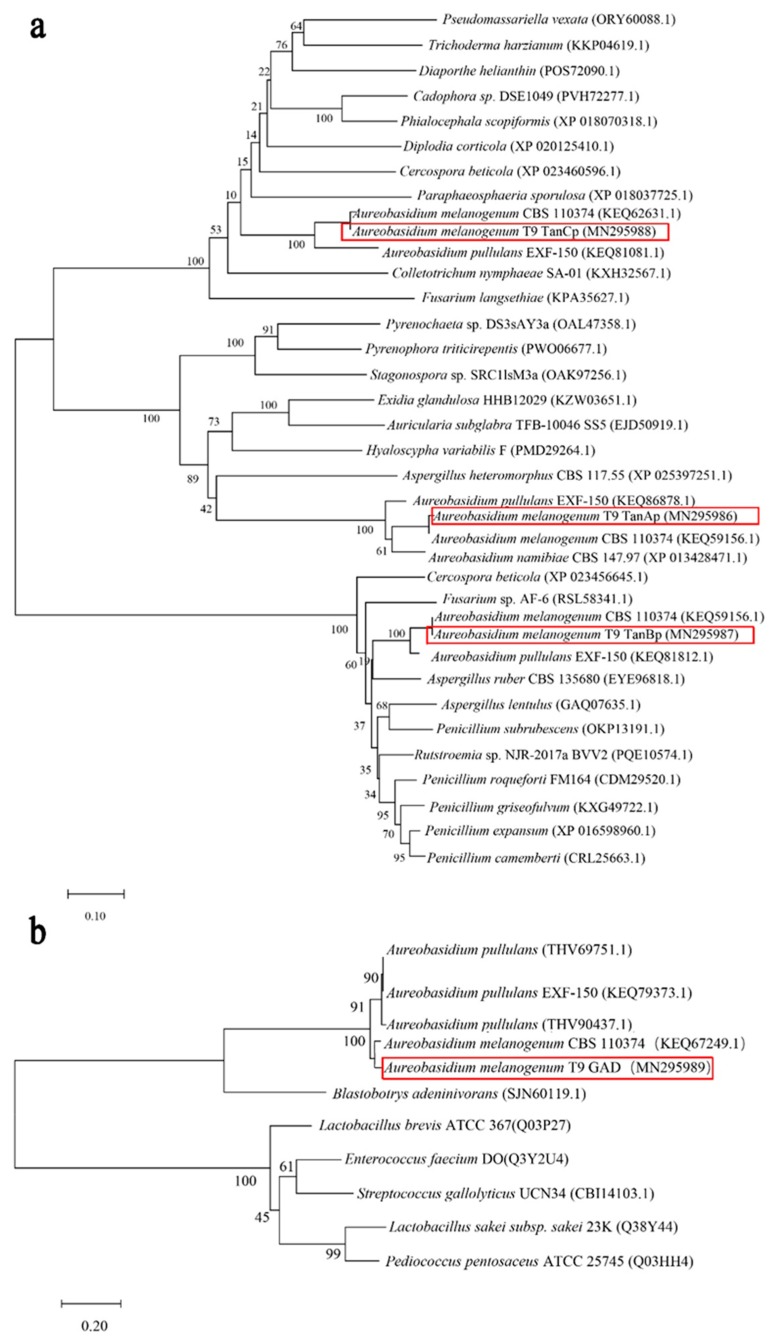
The phylogenetic tree of TanAp, TanBp and TanCp, and tannases from other yeasts and fungal species (**a**); the phylogenetic tree of GAD and gallic acid decarboxylases from other yeasts and fungal species (**b**).

**Figure 3 biomolecules-09-00439-f003:**
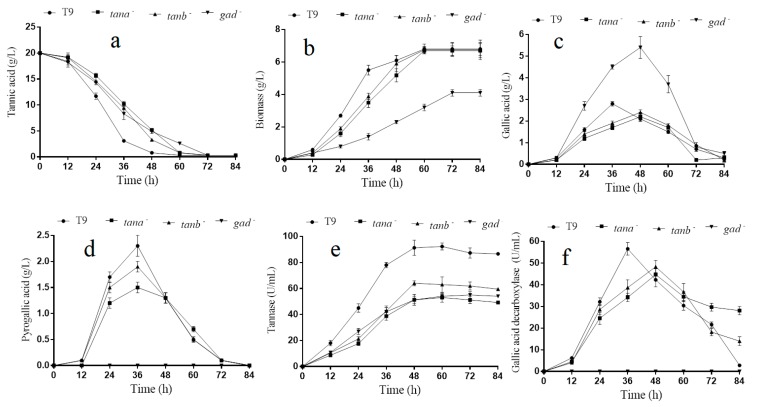
Tannic acid degradation by mutants and *A. melanogenum* T9 (**a**); biomass change of mutants and *A. melanogenum* T9 (**b**); gallic acid content change of mutants and *A. melanogenum* T9 (**c**); pyrogallic acid content change of mutants and *A. melanogenum* T9 (**d**); tannase activity change of mutants and *A. melanogenum* T9 (**e**); gallic acid decarboxylase activity change of mutants and *A. melanogenum* T9 (**f**). Data are given as means ± SD, *n* = 3.

**Table 1 biomolecules-09-00439-t001:** The primers used for qRT-PCR analyses in this study.

Primers	Sequence 5′-3′
TA5	CGATTGGAGCACCTTCAACGAGA
TA3	CGACCTGAAAGGGATGATGGGAT
TB5	TATTCGTGTTGTAGGTCGGGTCA
TB3	AGAACGGCACCATTACTGCTCAA′
TC5	GACCTCGACGTAACCAGACCTGA
TC3	CAACTGACGATGTTCCTTGCTCC
GD5	CAACAAGTTGGAAGCAAGGCAATA
GD3	CACAGCACGAGTGAGGTTGGGAT

**Table 2 biomolecules-09-00439-t002:** The primers used in this study. The underlined boxed bases are the shared bases.

Primers	Sequence 5′-3′
A5F	CCTGGCAACTCGTCCTACAACAT
A5R	GATCCCCCGAATTAGACCTGCATCTCCTTCAGTCCTT
A3F	ATGAGCCAACTGTCGCCGAGCCCTACGATTGGAGCACCTT
A3R	GGTTGAGTAGCGCCAGCGATGTA
B5F	GTCGATGGAAGCCTTGTCGTGTA
B5R	GATCCCCCGAATTACACTTATCCTGACCTGACCACCTT
B3F	ATGAGCCAACTGTCGAAAGGGAGAAACCACCTGGCAATT
B3R	TCCAACCAGCCATGAGTCACCTC
G5F	ATGAAGGTTCGCGAGATCTGTGAGG
G5R	GATCCCCCGAATTAATGTCGACTTGGGAGCCGATGATGC
G3F	ATGAGCCAACTGTCGAGATGGACAATGACGCCGACTGTCG
G3R	GATCATCCTCACCAGTCAAATCAGG
HPT5	TAATTCGGGGGATCTGGATTTTAGTACTGGA
HPT3	CGACAGTTGGCTCATCATCCGTTACATCA

**Table 3 biomolecules-09-00439-t003:** The property analysis of TanAp-, TanBp-, and TanCp-like tannases.

Protein	Accession Number	Strain	Residues (aa)	MW (KDa)	Signal Peptide
**TanAp-Like Proteins**					
Tannase	KEQ59156.1	*A. melanogenum* CBS 110374	528	57.23	Yes
Tannase	XP_013428471.1	*Aureobasidium namibiae* CBS 147.97	528	57.30	Yes
Tannase	KEQ86878.1	*Aureobasidium pullulans* EXF-150	542	55.15	No
Tannase	OAK97256.1	*Stagonospora* sp. SRC1lsM3a	538	56.57	No
Tannase	EJD50919.1	*Auricularia subglabra* TFB-10046 SS5	537	53.72	No
Tannase	PWO06677.1	*Pyrenophora triticirepentis*	540	55.97	No
**TanBp-like Proteins**					
Tannase	KEQ62357.1	*A. melanogenum* CBS 110374	587	63.39	No
Tannase	KEQ81812.1	*A. pullulans* EXF-150	583	62.95	No
Tannase	EYE96818.1	*Aspergillus ruber* CBS 135680	579	62.76	No
Tannase	KXG49722.1	*Penicillium griseofulvum*	580	62.85	Yes
Tannase	PQE10574.1	*Rutstroemia sp.* NJR-2017a BVV2	581	62.62	Yes
Tannase	CDM29520.1	*Penicillium roqueforti* FM164	579	62.57	Yes
Tannase	XP_023456645.1	*Cercospora beticola*	609	65.65	Yes
Tannase	XP_016598960.1	*Penicillium expansum*	580	62.83	Yes
Tannase	GAQ07635.1	*Aspergillus lentulus*	588	63.45	Yes
Tannase	CRL25663.1	*Penicillium camemberti*	580	62.86	Yes
Tannase	OKP13191.1	*Penicillium subrubescens*	589	64.00	No
Tannase	RSL58341.1	*Fusarium sp.* AF-6	581	63.20	Yes
**TanCp-like proteins**					
Tannase	KEQ62631.1	*A. melanogenum* CBS 110374	508	54.68	Yes
Tannase	KEQ81081.1	*A. pullulans* EXF-150	496	53.94	No
Tannase	XP_020125410.1	*Diplodia corticola*	542	58.16	No
Tannase	PVH72277.1	*Cadophora sp.* DSE1049	513	54.75	Yes
Tannase	XP_018037725.1	*Paraphaeosphaeria sporulosa*	508	55.17	Yes
Tannase	KXH32567.1	*Colletotrichum nymphaeae* SA-01	468	51.07	No
Tannase	XP_018070318.1	*Phialocephala scopiformis*	405	43.54	No
Tannase	ORY60088.1	*Pseudomassariella vexata*	451	49.84	No
Tannase	POS72090.1	*Diaporthe helianthi*	750	81.47	Yes
Tannase	KKP04619.1	*Trichoderma harzianum*	466	50.71	Yes
Tannase	KPA35627.1	*Fusarium langsethiae*	514	56.00	Yes

**Table 4 biomolecules-09-00439-t004:** The transcription level and translation level changes of the genes cultured in YPT medium compared to those cultured in YPD medium.

Gene	Protein	Gene Transcription Level Change (Fold)	Protein Translation Level Change (Fold)
*tanA*	TanAp	32.00 ± 3.6	8.22
*tanB*	TanBp	64.70 ± 5.2	332.00
*tanC*	TanCp	0.74 ± 0.2	-
*gad*	GAD	3.21 ± 0.3	-

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
