# Peer review of "Integrated Approaches to Reveal Genes Crucial for Tannin Degradation in Aureobasidium melanogenum T9"

_biomolecules, 2019, doi:10.3390/biom9090439_

Round 1

Reviewer 1 Report

Give the reference numbers of the microorganism strains and the collection from which they come.

Abstract:

Abstract is very poor, not clear for the reader and do not summarize the work. For instance: no introduction to the topic. It should be reformulated. Add more details, emphasize the importance of the results.

The article lacks a scientific description, characteristics and morphology of Aureobasidium strains

Please describe the properties of selenium as a compound with anti-oxidation properties in introduction. The language of the manuscript is not adequate for the desired journal level. The text may be handled by a native English speaker.

"Finally, the transcriptome analysis showed that tannic acid may have an adverse effect on the ribosome, oxidative phosphorylation biosynthesis of amino acids, and keep a low level of amino acid and proteins synthesis. This study provided new insights for the mechanism of tannins biodegradation by yeast. " - Let the authors give detailed results, because it is a bold statement. This is not supported by any result

Experimental section is documented unsatisfactorily. Headings and subtitles are unclear. State all instrumentation and chemicals/reagents used in a single subtitle. Describe analytical steps clearly for both techniques.

The statistics are unclear. Which tests were used.

Materials and methods section is incomplete and tables containing results should be improved (Headings and subtitles are unclear. State all instrumentation and chemicals/reagents used in a single subtitle. Describe analytical steps clearly for both techniques).

Most references are very old, give other examples.Some quotes are very badly done. Correct

Where in the article is "Conclusions" ???

"The transcriptome analysis showed that the most highly enriched pathways, such as fatty acid biosynthesis, carbon metabolism, and amino acid metabolism, are closely linked to the mechanisms of early tannic acid"-give details because it is only a statement of the authors. This is not a scientific breakthrough.

What is the industrial use of the results obtained and perspectives for the future - please write a short commentary in the article.

Reviewer 2 Report

Zhang et al. isolated Aureobasidium melanogenum strain T9 and analyzed single-gene disruptants of enzyme genes involved in tannin degradation. Overall, the manuscript is well-done and well-described. 

1. Tables 3 and 4 need more detailed explanation with notes. For examples, Accession number for what database (Table 3)? Transcription level changes between what samples (Table 4)? Although they are described in the main text and some of them are common sense for us (not need to explain), they should be clarified.

2. Do the author think double-gene disruption of tanA and tanB causes more significant defects in tannin degradation? Other tan gene(s) could be present in the genome of strain T9, which would be revealed by genome resequencing or RNA-seq data. Although I do not require to additional experiments and data analyses, please mention about these possibilities in Discussion in short.

3. Please show mapped reads number of each RNA-seq sample.

4. Line 209: Farias et al. 1992 is missing from reference list.

5. Line 386: Please add “under this culture condition”

6. Lines 389-391: in what species?

7. It would be better to move some descriptions in Results into Methods. For examples, Lines 221-224 and 279-280.

8. In my opinion, RNA-seq is not related to the current manuscript. I think the authors may not present this data in this manuscript. 

Author Response

Zhang et al. isolated Aureobasidium melanogenum strain T9 and analyzed single-gene disruptants of enzyme genes involved in tannin degradation. Overall, the manuscript is well-done and well-described. 

Authors’ response:

Thanks for your recognition.

Tables 3 and 4 need more detailed explanation with notes. For examples, Accession number for what database (Table 3)? Transcription level changes between what samples (Table 4)? Although they are described in the main text and some of them are common sense for us (not need to explain), they should be clarified.

Authors’ response:

  Thanks for the reviewer’s comments. Accession numbers used in this study were from NCBI website (https://www.ncbi.nlm.nih.gov/). The detailed explanation about Table 3 and table 4 were added into the revised manuscript. Sentences were added in Section 3.5 to clarify it.

Do the author think double-gene disruption of tanA and tanB causes more significant defects in tannin degradation? Other tan gene(s) could be present in the genome of strain T9, which would be revealed by genome resequencing or RNA-seq data. Although I do not require to additional experiments and data analyses, please mention about these possibilities in Discussion in short.

Authors’ response:

Thanks for your comments and suggestions. We think double-gene disruption of tanA and tanB would causes more significant defects in tannin degradation, as the transcriptional level of two genes were dramatically induced by tannin. Additional experiments were carried out to evaluate the transcriptional level analysis of tanA and tanB genes in different strains.

The mutation of any of those two genes induce a higher transcription of the other (Figure S2). Double-gene disruption of tanA and tanB would be carried out in future work, which may cause more significant defects in tannin degradation. The three tannase genes in this study were found based on the sequence similarity, and was confirmed by the RNA-seq data and label-free quantitative proteomics. However, in the RNA-seq data, some genes annotated with unknow functions have been found with certain degree of structural similarity with the identified tannases, which may be potential tannases. In future work, the functions of these genes would be identified.

The sentences were added in the fourth and fifith paragraphs of Discussion section.

Please show mapped reads number of each RNA-seq sample.

Authors’ response:

 We appreciate the reviewer’s comments. The average mapped reads number of RNA-seq from samples using glucose as the sole carbon source was 29037842 (97.94%). The average mapped reads number of RNA-seq from samples using tannic acid as the sole carbon source was 28289018 (97.55%). As suggested below, the transcriptome analysis section has been deleted in the new manuscript.

Line 209: Farias et al. 1992 is missing from reference list.

Authors’ response:

We appreciate the suggestion. The missing reference was added into the new manuscript.

Line 386: Please add “under this culture condition”

Authors’ response:

We appreciate the suggestion. This phrase has been   supplemented in the revised manuscript.

Lines 389-391: in what species?

Authors’ response:

Thanks for your comment. The yeast strain mentioned in the main text was Arxula adeninivorans and the missing sentence was added into the new manuscript.

It would be better to move some descriptions in Results into Methods. For examples, Lines 221-224 and 279-280.

Authors’ response:

We appreciate the reviewer’s suggestions and those words was moved into the Methods section.

In my opinion, RNA-seq is not related to the current manuscript. I think the authors may not present this data in this manuscript. 

Authors’ response:

As suggested by you, the RNA-seq data and analysis was removed from our new manuscript.

Reviewer 3 Report

The manuscript Manuscript ID: biomolecules-567061

Title: Integrated approaches to reveal genes crucial for tannin degradation in Aureobasidium melanogenum T9. Journal: Biomolecules.

Is an interesting work with many dates, however has several mistakes and need to be review careful.

Several sentences need to be review, for example.

Line 46 and 47: tannins can aggregate the precipitates in the beverages and raise serious environmental pollution problems. This sentence looks that all aggregates and precipitates in the beverage produce serious environmental pollutions. Is it so?

Line 137: These genes relative gene expression levels were analyzed using the comparative CT method. This sentence need to be review the writing style too.

The tables 3 and 4, are not mentioned in the text, just appear without any explanation.

In figure 3, the legend below the graph, the (e) is repeated and is missing the (f).

In line 397, “been associated with reduction of the tannase activity and tannic acid degradation rat during” appear the word “rat”, probably the authors want to say “rate”. Need to fix it.

In the result section appear information that correspond to discussion then, must be move it to that section.

According the discussion of the results, the authors need to discuss more about “tan A” and “tan B” can synergize between them or not, and if the mutation of any of those induce a higher transcription of the other.

Reviewer 4 Report

In this article authors described new strain of Aureobasidium melanogenum, T9 that has increased tolerance to tannins and higher performance for tannins degradation compared to other A. melanogenum strains. They characterized performance of this strain for tannins degradation and detected tan A, tan B, tan C and gad genes responsible for ability of this strain to degrade tannins. They also performed phylogenetic and transcriptome analysis to provide new insights for the mechanism of tannins degradation by yeast.

My opinion is that this article should be published after major revisions that are needed in order to connect results from this article more with literature data, and in order to emphasize novelty of this T9 strain for its ability to degrade tannins.

Major revision:

Since A. melanogenum T9 strain showed ability to degrade tannins and its tolerance was high compared to other A. melanogenum strains is this the first strain of A. melanogenum that can degrade tannins or there are other reported strains? This has to be clearly stated in the introduction section to emphasize the novelty of this strain (work).

Since A. melanogenum CBS 110374 strain is mentioned in the introduction section as the one with tannase activity than appropriate reference for this statement must be cited.

If there are other strains what is the performance of T9 strain compared to others especially A. melanogenum CBS 110374 strain reported in introduction section as the one with tannases activity?

Phylogenetic analysis shown on Figure 2 must be presented with higher resolution since it is hard to read names of the strains.

Why is A. melanogenum CBS 110374 labelled on Figure 2? There is no explanation for this in the text? Are the enzymes from this article the same as enzymes from A. melanogenum T9 strain?

Minor revision:

It is visually difficult to see concentration of tannic acid from Y axis and there is no need to show it since it can be read from X axis.

Round 2

Reviewer 1 Report

The article has been corrected. I accept the manuscript for publication

Author Response

We appreciate your acceptance of our revised manuscript. Thank for your all the comments and suggestions.

Reviewer 3 Report

The manuscript is improved and few mistakes must be corrected.

Author Response

We would like to thank you and all the expert reviewers for your kindly help to revise this manuscript and consider our manuscript for publication in Biomolecules after minor revisions. We appreciated all the comments and suggestions and carefully considered all of them during the revision. Here is a summary of revisions and responses.

Reviewer 4 Report

Authors replied to all of my comments.

Author Response

We would like to thanks for your all the comments and suggestions to help us kindly to revise this manuscript.